# How Did COVID-19 Impact the Antimicrobial Consumption and Bacterial Resistance Profiles in Brazil?

**DOI:** 10.3390/antibiotics12091374

**Published:** 2023-08-28

**Authors:** Natália Cassago Marcos Massarine, Gleyce Hellen de Almeida de Souza, Isadora Batista Nunes, Túlio Máximo Salomé, Marcelo dos Santos Barbosa, Izadora Faccin, Luana Rossato, Simone Simionatto

**Affiliations:** Health Sciences Research Laboratory, Universidade Federal da Grande Dourados, Dourados 79825-070, MS, Brazilgleycesouza@ufgd.edu.br (G.H.d.A.d.S.); isadora.nunes065@academico.ufgd.edu.br (I.B.N.); tuliomaximos@gmail.com (T.M.S.); marcelo_medvet@outlook.com (M.d.S.B.); izadora.faccin051@academico.ufgd.edu.br (I.F.); luanarossato@ufgd.edu.br (L.R.)

**Keywords:** SARS-CoV-2, healthcare-related infections (HAIs), antimicrobial resistance

## Abstract

The indiscriminate use of antibiotics has favored the selective pressure of multidrug resistance among microorganisms. This research evaluated the pattern of antibiotic prescriptions among the Brazilian population between January 2018 and December 2021. Additionally, the study sought to analyze the incidence rates of central line-associated bloodstream infection (CLABSI) and examine the profiles of antibiotic resistance. We assessed the hospital and community antimicrobial consumption from the National Health Surveillance Agency Database and correlated it to microorganisms. The consumption of antimicrobials in the hospital environment increased by 26% in 2021, highlighting polymyxin B, which increased by 204%. In 2021, 244,266 cases of CLABSI were reported, indicating a nosocomial infection rate of 7.9%. The rate of resistance to polymyxin B was higher in *Pseudomonas aeruginosa* (1400%) and *Klebsiella pneumoniae* (514%). Azithromycin emerged as the predominant antibiotic utilized within the community setting, accounting for 24% of the overall consumption. Pearson’s correlation analysis revealed a significant and positive correlation (r = 0.71) between the elevated usage of azithromycin and the incidence of COVID-19. Our results indicate an increase in antimicrobial consumption during the COVID-19 pandemic and reinforce the fact that the misuse of antimicrobials may lead to an expansion in antimicrobial resistance.

## 1. Introduction

Antimicrobial resistance (AMR) is a serious threat to human health in the 21st century globally [1]. Infections caused by multidrug resistance (MDR) microorganisms are associated with increased morbidity and mortality rates, length of stay, and treatment costs [2]. It was estimated that if there was no change in the global trend of antibiotic consumption, AMR could result in up to 10 million deaths by 2050 and cause a cumulative loss of at least USD 60 trillion in economic output [3]. Thus, the impact of AMR on public health and the economy in the short and medium term is enormous, particularly in developing countries such as Brazil [3,4].

The increase in the number of invasive procedures associated with the use of antibiotics, steroidal anti-inflammatory drugs, and other immunomodulatory drugs, as well as overcrowding in healthcare settings [5] during the COVID-19 pandemic, may contribute to the spread of MDR microorganisms. Despite the ineffectiveness of antibiotics in treating COVID-19, they were prescribed to individuals with suspected or confirmed cases of COVID-19 for several reasons. These reasons include the challenges in excluding bacterial coinfection during initial presentation and the potential occurrence of secondary bacterial infections throughout the course of the disease [6]. The empirical use of antibiotics for patients with COVID-19 raises concerns about their overuse and the subsequent harm associated with AMR [7]. Azithromycin (AZM) has been frequently prescribed in the treatment of COVID-19 globally [8].

In Brazil, an additional reason was the availability of the Brazilian COVID Kit in primary care for early treatment, which included AZM and other medications to reduce the transmission of the virus and, therefore, the spread of infection [8]. Although laws exist for restricting antibiotic use in Brazil, their implementation has remained insufficient. Moreover, most antibiotic stewardship programs primarily concentrate on hospitals where resistant infections are identified [9]. However, the majority of antibiotic consumption occurs in the community [9], representing up to 80% of the total consumption in several countries [10]. In certain countries, especially developing nations, the sale of antibiotics is not adequately regulated, resulting in their availability without the need for a prescription and therefore promoting self-medication [11]. The provision of antibiotics for human use in Brazil is strictly regulated, limiting access to individuals who possess a valid prescription from medical practitioners, dentists, or veterinarians. Since 2010, the enforcement of this regulation has involved monitoring the sales of these regulated substances [12,13]. To further enhance access to essential medications, the Brazilian government introduced the ‘Farmácia Popular’ Program (FPP). Under FPP, a specially selected list of medicines is subsidized by the government, and these medicines are made available at 517 public pharmacies spread across 440 municipalities. Additionally, individuals can also benefit from subsidized purchases at 34,627 registered private drugstores, which are located in 4467 municipalities [14,15].

Several countries, including Argentina, Uruguay, Ecuador, Guatemala, and Paraguay, have reported increased MDR infections due to the indiscriminate community use of antimicrobials during the pandemic [16]. This should serve as a warning to promote the rational use of antimicrobials, preventive measures, and multidisciplinary strategies for the prevention of healthcare-related infections (HAIs) [5,17]. The understanding of antibiotic consumption and the rise in AMR in Brazil during the COVID-19 pandemic is currently limited, highlighting a significant gap in knowledge that warrants further investigation and comprehension [18]. This study aimed to assess the trend in antibiotic prescriptions among the Brazilian population from January 2018 to December 2021, as well as to assess the differences in the rates of central line-associated bloodstream infection (CLABSI) and resistance profiles of the identified bacteria.

## 2. Results

### 2.1. Antimicrobial Consumption in Hospitals

According to the data mapping conducted in January 2021, based on the National Registry of Health Establishments of the Computer Department of the Unified Health System (DATASUS), the NHSA, and the Brazilian Institute of Geography and Statistics, Brazil has a total of 45,848 ICU beds. Of these, 22,844 ICU beds are part of the Unified Health System (SUS), while 23,004 ICU beds are associated with the private healthcare system. The percentage of registered ICUs that disclosed their data during each year of the study was 57%, 45%, 54%, and 74% in 2018, 2019, 2020, and 2021, respectively.

The average consumption of antimicrobials for hospital use, expressed in DDD/1000 beds-day, was 26% higher in 2021 than in 2019. The main contributors to this increase were PMB (204%), CRO (12%), PIP (10%), and MEM (38%). The consumption of CIP exhibited a notable decline of 44% (20.21 g vs. 11.29 g). While MEM was the predominant antibiotic in terms of consumption in both 2018 and 2019, the consumption of CRO surpassed that of MEM in 2020. Although MEM continued to be the primary antibiotic in terms of consumption during 2018 and 2019, there was a notable shift in 2020, with CRO consumption surpassing that of MEM. However, it is worth highlighting that this trend was reversed in 2021, as CRO consumption declined, while MEM consumption increased.

The consumption of CRO showed a significant increase of 56% from 2018 to 2020. However, in 2021, this increasing trend did not continue; instead, there was a reduction of −28% in CRO consumption. In 2021, there was an increase in the consumption of MEM (10%) and PMB (48%), while the others remained unchanged (Figure 1).

The number of patients with device days was higher after the start of the COVID-19 pandemic, increasing the incidence of CLABSI. The incidence density of CLABSI demonstrated an increasing pattern from 2018 to 2021, with rates of 4.1, 3.9, 4.3, and 5.2, respectively, for each respective year. In 2020, a total of 41,600 CLABSI cases were reported by hospitals in Brazil, which indicate a nosocomial infection rate (number of HAI episodes/total number of patients admitted × 100) of 1.34%. In 2021, a total of 244,266 cases of CLABSI cases were documented, leading to a nosocomial infection rate of 7.9%. This rate is higher than the rates observed in previous years.

During the study period, 110,217 strains were tested for their resistance profile for the following antibiotics: oxacillin and VAN for Gram-positive bacteria and carbapenems, PMB, and third- and fourth-generation cephalosporin for Gram-negative bacteria (Figure 2). Our data showed that, during the study period, *Klebsiella pneumoniae* (20%), *Pseudomonas aeruginosa* (7.5%), and *Acinetobacter* (9.6%) were the most common Gram-negative bacteria. Among the Gram-positive bacteria, coagulase-negative *Staphylococcus* spp. (31%), *Staphylococcus aureus* (13.5%), and *Enterococcus faecalis* (5%) were the most common.

On the other hand, the resistance profile was not the same over the years. The levels of bacterial resistance observed in CLABSI cases within ICU settings exhibited a significant increase during 2020 and 2021, coinciding with the occurrence of the COVID-19 pandemic. Notably, there was a substantial rise in resistance to oxacillin among coagulase-negative *Staphylococcus* species, with a 41% higher resistance rate observed after the onset of the pandemic compared with the pre-pandemic period. Regarding the Gram-negative bacteria, the resistance to PMB was higher in 2021, for all the strains tested. The resistance rates of *P*. *aeruginosa* and *K. pneumoniae* increased by 1400% and 514% in 2021, respectively (Figure 2). The CLABSI data from adult intensive care units for the period from 2018 to 2021 can be seen in Figure A1.

### 2.2. Antimicrobial Consumption in the Community

Throughout the entire study period, the Brazilian community consumed a total of 292,943,776 boxes of antibiotics. A comprehensive analysis was conducted on a total of 187,100,807 prescriptions issued by medical professionals, veterinarians, and dentists. Among these prescriptions, 23,336,647 (12%) were issued by veterinarians, 3,399,002 (0.2%) were issued by dentists, and the majority were issued by medical practitioners. The most consumed antibiotic was AZM, which contributed to 23.5% (68,846,570 boxes) of the total consumption, followed by AMX (20.7%) and LEX (20.6%). ERY (0.3%) and CLR (1.7%) were the least consumed antibiotics during the study period. The other antibiotics, namely, TET, TMP, SMX, LVX, and CIP, accounted for the remaining 33% (53,364,135 boxes) of the total consumption.

The community consumption of AZM was analyzed, and five consumption peaks were observed from January 2017 to December 2021. The first peak occurred in the post-pandemic period (June 2020), when 912.40 boxes/100,000 inhabitants were consumed. The second was in December 2020, when 1305 boxes/100,000 inhabitants were consumed, and the third was in March 2021, when 1120.81 boxes/100,000 inhabitants were consumed. The fourth peak occurred in July 2021, with a consumption rate of 1126.90 boxes/100,000 inhabitants. The final and largest peak was observed in September 2021, with a consumption rate of 1614.61 boxes/100,000 inhabitants (Figure 3).

The average consumption of AZM before and after the onset of COVID-19 in Brazil changed. Before the onset of COVID-19, the mean consumption of AZM was 12.844 boxes/100,000 inhabitants, which increased to 18.392 boxes/100,000 inhabitants, representing a 43% increase. The COVID-19 cases were examined to determine if there was any association with an upsurge in the consumption of AZM. In a nonempirical evaluation of the graphs of AZM consumption and COVID-19 cases, an alignment between the peaks and valleys of the two variables was observed, while Pearson’s (*p* = 0.717 (95% CI: 0.543–0.831)) and Spearman’s (s = 0.720 (95% CI: 0.542–0.836)) correlation coefficients indicated moderate-to-strong associations (Figure 3).

We compared the increase in the number of COVID-19 cases with the consumption of other antibiotics, such as CIP and LEX, and did not observe an association between them, as seen with AZM. On the other hand, the mean consumption of LEX and CIP decreased after the onset of COVID-19. Furthermore, the correlation analysis did not indicate a correlation between the consumption of LEX or CIP and the onset of COVID-19 or between the consumption of the two antibiotics.

## 3. Discussion

The indiscriminate use of antimicrobials contributes to the increasing rates of AMR; thus, evidence of antimicrobial consumption may help design an AMR scenario [19]. Our research aimed to examine the patterns of antimicrobial consumption in Brazil both before and during the COVID-19 pandemic. All the different analyses conducted revealed a strong relationship between the increase in antibiotic consumption and COVID-19 cases.

The utilization of hospital antibiotics, measured in terms of DDD, exhibited a 26% increase in 2021 when compared with 2019. Among the antibiotics consumed, MEM was identified as the most frequently utilized. A higher overall antimicrobial use has been reported in COVID-19 patients (74.6% and 72% of total COVID-19 patients) [20,21], from China [22]. In a smaller Brazilian cohort of 72 hospitalized patients, 84.7% received antibiotic therapy [23]. The high percentage of antimicrobials contrasts with the low incidence of coinfections and secondary infections in COVID-19 patients reported in the literature. The overall rate of bacterial infections was 7% [24].

Antibiotic consumption in Brazil is consistent with the published literature, in which treatment with cephalosporins, fluoroquinolones, macrolides, beta-lactams with beta-lactamase inhibitors, and carbapenems are the most commonly described [20,21,23]. Our study revealed a striking 204% increase in the use of PMB, raising significant concerns due to the limited treatment alternatives available. Patients who had undergone surgical procedures or had prior exposure to carbapenems were found to be at a 16.5 and 45.5 times higher risk, respectively, of developing an infection resistant to polymyxins [25]. The occurrence of PMB resistance is associated with an increased risk of hospital mortality [26,27] and deserves attention. Even before the pandemic, Brazil had a high resistance rate to PMB of 27.1% [28]. After the pandemic, these numbers increased, mainly for *P. aeruginosa* and *K. pneumoniae* (30% and 43%, respectively). PMB is currently considered the last line of defense against Gram-negative bacteria, notably carbapenem-resistant Enterobacteriaceae, *P. aeruginosa*, and *A. baumannii*, which have been classified as urgent or serious threats by the US Centers for Disease Control and Prevention (CDC) [29]. On the other hand, carbapenems and broad-spectrum resistance among *Klebsiella pneumoniae* were the most frequent in Italy, with resistance rates of 65–84% in ICU samples [30]. During the pandemic, the incidence of carbapenem-resistant enterobacterial colonization in Italy increased from 6.7% in 2019 to 50% in March 2020 [31].

Several reports have described an increase in MDR organisms during the COVID-19 pandemic [21,31,32]. In previous pandemics, such as the 2003 SARS outbreak, an escalation in antimicrobial resistance was also documented, with Methicillin-resistant *Staphylococcus aureus* (MRSA) exhibiting increased prevalence during that period. A similar trend was observed during the Middle East respiratory syndrome (MERS) outbreak, where MRSA and carbapenem-resistant *Acinetobacter baumannii* were the most frequently encountered bacteria [33]. In our study, a total of 244,266 cases of CLABSI were reported during the second year of the pandemic, reflecting a substantial 170% increase in 2021 when compared to 2018. In Brazil, the high prevalence of HAI caused by multi resistant microorganisms is particularly worrying [5]. In the US, 1.7 million HAIs are reported per year, with a mortality rate of 99,000 per year, and the cost for treating resistant infections reaching USD 10 billion [2]. Coagulase-negative *Staphylococcus* (31%), *K. pneumoniae* (20%), *S. aureus* (13.5%), *A. baumannii* (9.6%), and *P. aeruginosa* (7.5%) were the most commonly isolated strains in our study. These findings were in line with studies in Italian ICUs from 2015 to 2019 [30], where the predominance of Gram-positive pathogens was documented [34,35].

Furthermore, the results of community-consumed antibiotics in this study indicate that AZM, AMX, and LEX were the most consumed antibiotics, which corroborates with the pattern in the period from 2013 to 2016 [36] and 2008 to 2012 [37]. Moreover, these antibiotics were the most consumed by the community in the European Union from 1997 to 2017 [19]. Brazil showed an increase in AZM consumption during the pandemic period, similar to the trend reported in 10 African countries: Ghana, Kenya, Uganda, Nigeria, South Africa, Zimbabwe, Botswana, Liberia, Ethiopia, and Rwanda [38]. However, other countries such as the European Union [39], Canada [40], and the US [41] reported a decrease in the community consumption of antibiotics.

During the COVID-19 pandemic, Canada showed a 26.5% decrease in antibiotic consumption by community retail pharmacies compared with 2019 [42]. This decrease was 70% in children and 34% in the elderly [42]. Before the COVID-19 pandemic, our study demonstrated a consistent pattern of antibiotic consumption, with AMX being the most frequently prescribed antibiotic, followed by LEX and AZM. However, in 2020, a shift in consumption patterns was observed, and AZM emerged as the predominant antibiotic prescribed, likely attributed to the onset of the COVID-19 pandemic on 30 January 2020 [6]. This was confirmed by Pearson’s (*p* = 0.71) and Spearman’s (s = 0.72) correlation coefficients, both indicating a strong correlation between the increase in COVID-19 cases and the consumption of AZM. In contrast, LEX and CIP showed a weak correlation. Both AMX and AZM have been widely used against respiratory infections, which occur more frequently during the coldest months of the year [43]. AZM and AMX, which are used as the first choice for respiratory infections, accounted for half of the antibiotics prescribed in the country [44].

The WHO recommends the use of AZM in the treatment of moderate-to-severe COVID-19 with a risk of bacterial infection [6], which probably encouraged the increase in community antimicrobial consumption, supposedly for mild cases [42,43]. The availability of the Brazilian COVID Kit, which includes combinations of chloroquine/hydroxychloroquine, AZM, and ivermectin, as a treatment option for initial COVID-19 symptoms, has contributed to the rise in the community consumption of AZM. In Brazil, it was widely distributed to the general population in primary care [8]. It is estimated that one in four Brazilians have consumed drugs from the COVID Kit [8]. Moreover, data from the DETECTCoV-19 study [8,45] showed that 77% of people with a previous diagnosis of COVID-19 used AZM [35,39]. This possible selective pressure imposed by the exacerbated use of AZM may favor microorganisms causing lower respiratory tract infections, such as bronchitis and pneumonia, and upper respiratory tract infections, such as sinusitis, pharyngitis, and otitis [46].

This high rate of antimicrobial consumption in the treatment of COVID-19 may be related to the symptoms being similar to bacterial pneumonia, the fear regarding the magnitude of the pandemic and the number of deaths, the uncertainty in the association of coinfections or secondary infections, and the lack of effective treatments and protocols [47]. Furthermore, the nonmandatory reporting of antibiotic consumption in Brazilian hospitals introduces potential information bias. The number of hospitals reporting these data fluctuates on a monthly basis, creating challenges in accurately comparing the rates of antibiotic consumption. Although the major strength of this study is the completeness of the recorded data, the aggregated nature of the collected data is a limitation. The data only contained information on antibiotic consumption rates in the country. We did not segregate the findings based on age (e.g., children or elderly) or sex. Moreover, the aggregation of the data failed to distinguish between appropriate and inappropriate antibiotic prescriptions within the high rate of antibiotic consumption, which is crucial to fighting the overconsumption of antibiotics. The potentially high use of antimicrobials in COVID-19 patients may also lead to a shift from an increase in short-term COVID-19 mortality to an increase in long-term mortality, i.e., AMR mortality, with our biggest concern being the emergence of a new pandemic-related bacterial resistance [42]. Thus, managing the use of antimicrobials will continue to be an administrative challenge for Brazilian health services.

## 4. Materials and Methods

### 4.1. Study Design

We performed a cross-sectional, observational, and retrospective study of antibiotic consumption registered by the National Health Surveillance Agency (NHSA) from 2018 to December 2021 based on hospital and community consumption in Brazil. Brazil has 213.3 million inhabitants and is divided into 26 states, and federal districts with 5570 municipalities in 2022 [48]. Since the data utilized in this study were obtained from open sources, there was no requirement for approval from the research ethics committee.

#### Hospital Setting

To achieve a comprehensive understanding of antibiotic consumption, an analysis of pharmacy records encompassing all antimicrobials administered to hospitalized patients was conducted. In Brazil, the NHSA recommends that hospitals collect and report data on HAIs using FormSus version 3.0 [49]. However, this information is not mandatory, and the number of hospitals reporting this information varies every month [49]. The data are public and available on the NHSA website (https://centralpaineis.cgu.gov.br/visualizar/dadosabertos, accessed on 2 January 2021). To ensure the integrity of the data, only hospitals that reported CLABSI cases for a minimum of 10 to 12 months per year were included in the analysis. CLABSI is defined as a laboratory-confirmed bloodstream infection occurring in a patient with a central venous catheter in place within 48 h before the onset of infection, excluding infections originating from another site [49]. The incidence rates and phenotypic profiles of CLABSI in all adults in intensive care units (ICUs) were examined for the period from January 2018 to December 2021. The CLABSI indicators were calculated as percentages based on aggregate data for the specified period. The ICU indicators were presented as incidence density (DI) using the following equation (Equation (1)):DI = *the sum of the number of central line-associated bloodstream infections/sum of patients with the invasive device per day* × 1000.(1)

### 4.2. Antibiotics

For analysis of hospital antimicrobial consumption, the following categories of antimicrobials were selected: third or fourth generation of cephalosporin (ceftriaxone (CRO) and cefepime (FEP)), fluoroquinolones (ciprofloxacin (CIP)), carbapenems (meropenem (MEM)), penicillin (piperacillin (PIP)), polypeptides (polymyxin (PMB)), and glycopeptides (vancomycin (VAN)). The selection of these categories was based on their frequent utilization within hospital settings, their wide-ranging efficacy, and their direct relevance to clinical practice. These choices are in accordance with ANVISA’s recommended protocols for antimicrobial usage.

Based on the data from Intercontinental Medical Statistics Health Brazil (IMS Health Brazil), it was observed that six classes and ten specific substances comprised approximately 96% of community sales in Brazil. Based on this finding, we included the following six classes beta-lactams (amoxicillin (AMX) and cephalexin (LEX)), pyridines (trimethoprim (TMP)), sulfonamides (sulfamethoxazole (SMX)), tetracycline (tetracycline (TET)), macrolides (azithromycin (AZM), clarithromycin (CLR), and erythromycin (ERY)), and fluoroquinolones (ciprofloxacin (CIP) and levofloxacin (LVX)) [14]. The drugs administered through routes other than parenteral, specifically oral antibiotics, were excluded from this study. This decision was made because, in Brazil, patients are seldom hospitalized in the ICU solely for oral antibiotic treatment. Consequently, this figure had minimal significance and was excluded from the analysis.

### 4.3. Data Analysis

#### 4.3.1. Defined Daily Dose—DDD

In Brazil, the NHSA offers an electronic form to report the estimated consumption of the key antimicrobials utilized in ICUs. The Defined Daily Dose (DDD) is a standardized measurement that represents the estimated average maintenance dose per day for a drug when administered for its primary therapeutic indication in adults weighing 70 kg [50]. The DDD values are based on the Anatomical Therapeutic Chemical (ATC)/DDD index of the year 2020 or the most recent available version. The consumption of antimicrobials was verified in DDD/1000 patient days, according to Equation (2).
DDD/1000 patient days= *(total antimicrobial consumed in grams, in the month of surveillance)/(DDD established for the drug)* × *(Patient days, in the month of surveillance).*(2)

To investigate whether there was an increase in antimicrobial consumption during the pandemic in comparison to the preceding two years, the data retrieved from the open data survey of the NHSA system were examined. The analysis involved the use of tables prepared using GraphPad Prism software (San Diego, CA, USA) version 7, and the results are presented as graphs and descriptive statistics.

#### 4.3.2. The Community Consumption of AZM, LEX, and CIP in Relation to the Occurrence of COVID-19

In order to analyze the impact of the pandemic on antimicrobial consumption, the patterns of three specific antibiotics were compared: AZM, LEX, and CIP. AZM was selected because it was used in the treatment of COVID-19, while LEX and CIP were chosen because they had a high consumption rate in Brazil before the pandemic. Consequently, a correlation analysis was performed to examine the relationship between the monthly count of COVID-19 cases and the corresponding rise in antibiotic usage. The data pertaining to the consumption of AZM, CIP, and LEX were collected from 2017 to 2021 and adjusted per 100,000 inhabitants, utilizing the population estimates provided by the Brazilian Institute of Geography and Statistics [51]. Correlation analyses were performed using the software GraphPad Prism version 7. To accurately analyze the seasonal peak of antibiotic consumption, a specific approach was implemented. The period from January to February 2020, when COVID-19 cases began to be reported in Brazil, was assigned the value of zero for both the number of cases and antibiotic consumption. An adjustment was implemented to account for the initial phase of the pandemic. Two statistical analyses were performed to identify the seasonal peak of antibiotic consumption. These analyses aimed to assess the level of correlation between two variables: the number of COVID-19 cases and the consumption of AZM. By examining the correlation between these variables, it becomes feasible to discern any patterns or connections between the escalation in COVID-19 cases and the accompanying increase in AZM consumption during specific time intervals. Pearson’s and Spearman’s correlation coefficients were determined using the statistical software IBM Statistical Package, version 21 for the Social Science (located in Chicago, IL, USA), whereas, for non-normally distributed data, the natural log was used.

## 5. Conclusions

The results indicate that there was an increase in antimicrobial consumption during the COVID-19 pandemic in Brazil. The indiscriminate use of antibiotics has led to an increase in AMR, and we need to review the actions taken. Health authorities must implement measures to mitigate the inappropriate prescription of antibiotics, reinstate control measures, and allocate resources to enhance accurate diagnoses. These actions are essential for curbing the dissemination of AMR and preparing for the potential emergence of the next silent pandemic, the AMR pandemic.

## Figures and Tables

**Figure 1 antibiotics-12-01374-f001:**
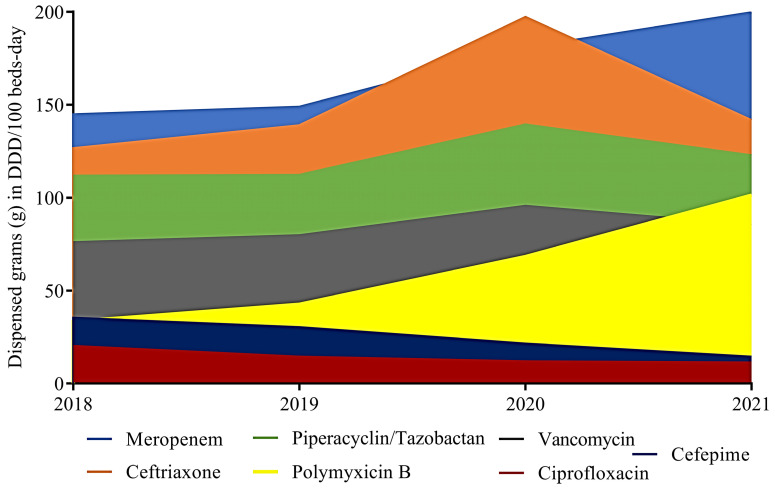
The average antibiotic consumption in the Brazilian hospital environment, expressed in DDD/1000 beds-day.

**Figure 2 antibiotics-12-01374-f002:**
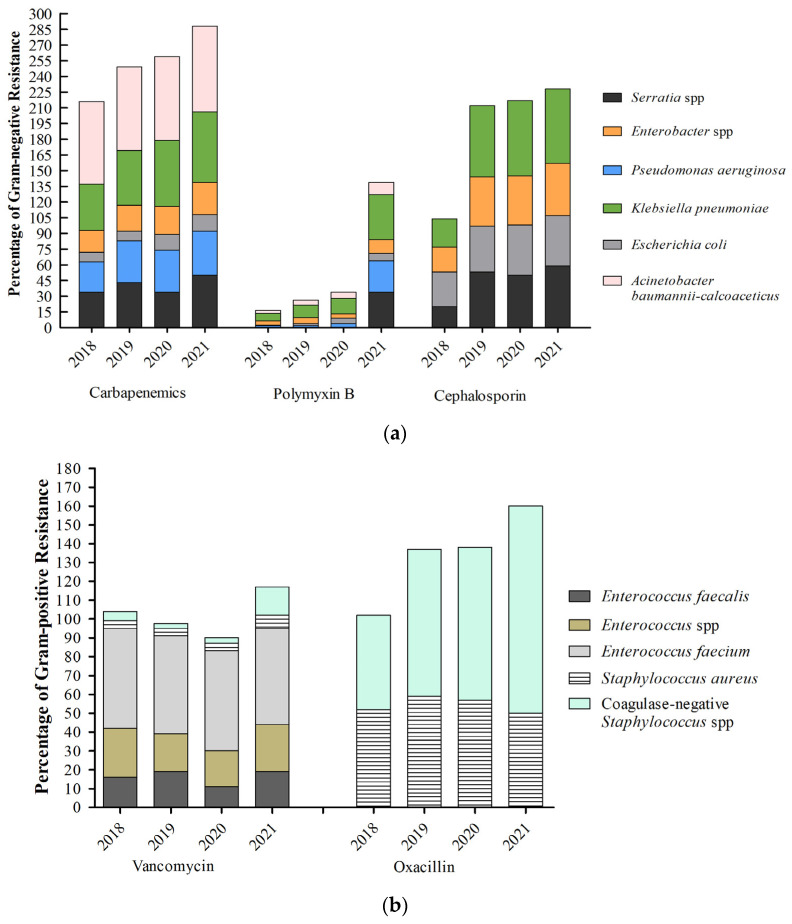
Antimicrobial resistance profile of Gram-negative (**a**) and Gram-positive (**b**) bacteria in patients admitted to ICUs in Brazil, with primary bloodstream infections associated with the central venous catheter. The described data are from adults admitted to the intensive care unit and are expressed as a percentage from 2018 to 2021.

**Figure 3 antibiotics-12-01374-f003:**
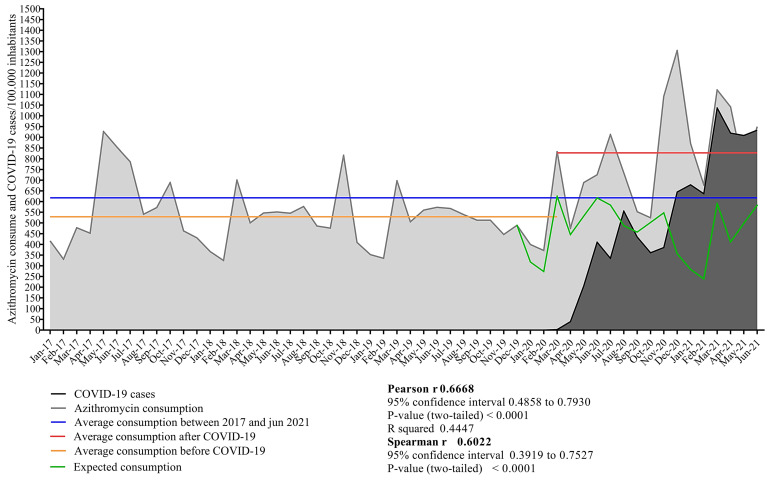
The correlation between the community consumption of azithromycin and COVID-19 cases in 100,000 inhabitants in Brazil from January 2017 to December 2021. The average consumption of azithromycin was evaluated both before and after the emergence of COVID-19 cases in Brazil, along with a consumption forecast based on the pre-pandemic period. The red line represents the average consumption after COVID-19, the blue line represents the average consumption between 2017 and 2021, the orange line represents the average consumption before COVID-19, and the black line represents the projected consumption.

## Data Availability

The datasets generated during and/or analyzed during the current study are available from the corresponding author upon reasonable request.

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
