# Peer review of "How Did COVID-19 Impact the Antimicrobial Consumption and Bacterial Resistance Profiles in Brazil?"

_antibiotics, 2023, doi:10.3390/antibiotics12091374_

Round 1
Reviewer 1 Report
The manuscript is generally well-written and the topic is very interesting and timely.
I have the following suggestions for the authors:
Please delete the first sentence in the Introduction (line 32).
Page 2, line 52, "an additional reason" instead to just "an additional"
Page 2, line 65. What is the meaning of "The Brazilian government maintains ...", please explain it more clearly.
Figure 1, legend. I suggest adding the abbreviations in brackets after the names of antibiotics, to be better understood in line with the text.
Page 8, line 320. What is the reason for selecting these particular categories of antimicrobials for hospital consumption? Such a reason is given later in this paragraph for the selection of antimicrobials in the community; therefore, it should be mentioned for the hospital setting too.
line 327. six classes:
Page 7, line 243. Correct: other countries such as the European Union
Limitations of the study are appropriately addressed.
Minor corrections needed.
Author Response
18th August 2023.
Thank you very much for inviting us to submit a revised version of our manuscript ID: antibiotics-2565036, entitled: "How did COVID-19 impact the antimicrobials consumption and bacterial resistance profiles in Brazil?" for publication in the Antibiotics. Indeed, the reviewers have raised a number of important concerns. We have now made a thorough revision of the manuscript taking into account these points, which helped in improving the quality of the manuscript. Please find bellow a point-by-point response to the reviewers and their comments.
REVIEWERS COMMENTS
REVIEWER #1:
The manuscript is generally well-written and the topic is very interesting and timely. I have the following suggestions for the authors:
- Please delete the first sentence in the Introduction (line 32).
Response: Thank you for your feedback on our manuscript. We appreciate the reviewers’ constructive comments and apologize for the confusion. The first sentence of the Introduction has been removed.
- Page 2, line 52, "an additional reason" instead to just "an additional".
Response: The sentence was revised following the reviewer’s suggestion (Page 2; line 51).
“In Brazil, an additional reason was the availability of the Brazilian Covid Kit in primary care for early treatment, which included AZM and other medications to reduce the transmission of the virus and, therefore, the spread of infection [8].”
- Page 2, line 65. What is the meaning of "The Brazilian government maintains ...", please explain it more clearly.
Response: We appreciate your valuable input, and our revised manuscript includes a clearer statement about this Brazilian government-subsidized program to expand the population's access to essential medicines (Page 2; lines 63-69). We believe this explanation offers a clearer understanding of the context.
“To further enhance access to essential medications, the Brazilian government introduced the 'Farmácia Popular' Program (FPP). Under FPP, a specially selected list of medicines is subsidized by the government, and these medicines are made available at 517 public pharmacies spread across 440 municipalities. Additionally, individuals can also benefit from subsidized purchases at 34,627 registered private drugstores, which are located in 4,467 municipalities”.
- Figure 1, legend. I suggest adding the abbreviations in brackets after the names of antibiotics, to be better understood in line with the text.
Response: Thanks to the reviewer for this valuable suggestion. The legend for Figure 1 has been revised as recommended, incorporating the abbreviations corresponding to the antibiotics. This adjustment aims to enhance the alignment between the figure and the corresponding text.
- Page 8, line 320. What is the reason for selecting these particular categories of antimicrobials for hospital consumption? Such a reason is given later in this paragraph for the selection of antimicrobials in the community; therefore, it should be mentioned for the hospital setting too.
Response: In response to your comment, we recognize the importance of providing specific data on the rationale behind the selection of specific categories of antimicrobials for hospital use. The manuscript has been revised to provide an explanation (Page 9; lines 332-335).
“The selection of these categories was based on their frequent utilization within hospital settings, their wide-ranging efficacy, and their direct relevance to clinical practice. These choices are in accordance with ANVISA's recommended protocols for antimicrobial usage.”
- line 327. six classes:
Response: The sentence was revised following the reviewer’s suggestion (Page 9; line 337).
“Based on data from Intercontinental Medical Statistics Health Brazil (IMS Health Brazil), it was observed that six classes and ten specific substances comprised approximately 96% of community sales in Brazil.”
- Page 7, line 243. Correct: other countries such as the European Union
Response: The sentence was revised following the reviewer’s suggestion (Page 8; line 252).
“Brazil showed an increase in AZM consumption during the pandemic period, similar to the trend reported in 10 African countries: Ghana, Kenya, Uganda, Nigeria, South Africa, Zimbabwe, Botswana, Liberia, Ethiopia, and Rwanda [38]. However, other countries such as the European Union [39], Canada [40] and the US [42] reported a decrease in community antibiotic consumption.”
- Limitations of the study are appropriately addressed.
Response: We appreciate the reviewer's comment and are pleased to have adequately addressed this aspect. Thank you for your time and expertise in reviewing our work.
Reviewer 2 Report
The authors investigated the effects of COVID-19 pandemic to national antimicrobial consumption changes in Brazil. Antimicrobial resistance profile changes of major pathogenic bacteria for selected drugs also were presented. This article will provide valuable information to manage the endemic stage of the COVID-19. Please add a figure showing antimicrobial community consumption. antimicrobial community consumption has been greatly affected by the COVID-19 in Brazil. A similar graph as in Figure 1 for community consumption will be great help.
Minor points:
Figure 1. It is hard to discern some of the data. Only partial information is available to be seen for Ceftriaxone and Polymyxin B. Please change graph format.
Line 32-33. Please delete the first sentence.
Line 52. Are some words missed after “an additional”?
Line 94-95. the upward trend not to be seen at Figure 1.
Line 216 – 218. The reference 28 doesn’t contain data during the pandemic. The only the samples collected 2015-2019 were examined in the paper. Please cite the correct reference.
Line 242-244, “European Union,[39] Canada,[40] the US,[41] and the United Kingdom[38]” should be change to “European Union (need reference), Canada[39], the US [40], and the United Kingdom (need correct reference)”
Please check all the citation for any errors.
Author Response
18th August 2023.
Thank you very much for inviting us to submit a revised version of our manuscript ID: antibiotics-2565036, entitled: "How did COVID-19 impact the antimicrobials consumption and bacterial resistance profiles in Brazil?" for publication in the Antibiotics. Indeed, the reviewers have raised a number of important concerns. We have now made a thorough revision of the manuscript taking into account these points, which helped in improving the quality of the manuscript. Please find bellow a point-by-point response to the reviewers and their comments.
REVIEWER #2:
The authors investigated the effects of COVID-19 pandemic to national antimicrobial consumption changes in Brazil. Antimicrobial resistance profile changes of major pathogenic bacteria for selected drugs also were presented. This article will provide valuable information to manage the endemic stage of the COVID-19. Please add a figure showing antimicrobial community consumption. antimicrobial community consumption has been greatly affected by the COVID-19 in Brazil. A similar graph as in Figure 1 for community consumption will be great help.
Minor points:
- Figure 1. It is hard to discern some of the data. Only partial information is available to be seen for Ceftriaxone and Polymyxin B. Please change graph format.
Response: Thank you for your valuable contribution to enhancing our research. In response to your suggestion, we will be revising the graph format of Figure 1 to enhance data visibility, particularly for Ceftriaxone and Polymyxin B.
- Line 32-33. Please delete the first sentence.
Response: We appreciate the reviewers’ constructive comments and apologize for the confusion. The first sentence of the Introduction has been removed.
- Line 52. Are some words missed after “an additional”?
Response: The sentence was revised following the reviewer’s suggestion (Page 2; line 51).
“In Brazil, an additional reason was the availability of the Brazilian Covid Kit in primary care for early treatment, which included AZM and other medications to reduce the transmission of the virus and, therefore, the spread of infection [8].”
- Line 94-95. the upward trend not to be seen at Figure 1.
Response: Thank you for bringing this to our attention. We apologize for the confusion. The sentence has been revised in accordance with the reviewer's feedback (Page 2; line 97-100)."
“Although MEM continued to be the primary antibiotic in terms of consumption during 2018 and 2019, there was a notable shift in 2020, with CRO consumption surpassing that of MEM. However, it's worth highlighting that this trend was reversed in 2021, as CRO consumption declined while MEM consumption growth.”
- Line 216 – 218. The reference 28 doesn’t contain data during the pandemic. The only the samples collected 2015-2019 were examined in the paper. Please cite the correct reference.
Response: We appreciate your attention to detail. We apologize for the confusion. We revised the sentence and corrected the reference citation.
“On the other hand, carbapenems and broad-spectrum resistance among Klebsiella pneumoniae were the most frequent in Italy, with resistance rates of 65-84% in ICU samples [30]. During the pandemic, the incidence of carbapenem-resistant enterobacterial colonization in Italy increased from 6.7% in 2019 to 50% in March 2020 [31]”.
- Line 242-244, “European Union,[39] Canada,[40] the US,[41] and the United Kingdom[38]” should be change to “European Union (need reference), Canada[39], the US [40], and the United Kingdom (need correct reference)”
Response: The sentence was revised following the reviewer’s suggestion (Page 8; lines 252-254).
“Brazil showed an increase in AZM consumption during the pandemic period, similar to the trend reported in 10 African countries: Ghana, Kenya, Uganda, Nigeria, South Africa, Zimbabwe, Botswana, Liberia, Ethiopia, and Rwanda [38]. However, other countries such as the European Union [39], Canada [40] and the US [42] reported a decrease in community antibiotic consumption.”
- Please check all the citation for any errors.
Response: We appreciate your diligent review and valuable contribution. Thanks for the fixes you provided. We will do a comprehensive review of all quotes to ensure their accuracy and consistency. Thank you for your time and expertise in reviewing our work.
Your Sincerely,
Authors of the manuscript